# Optimizing plastic film mulch to improve the yield and water use efficiency of dryland maize in the Loess Plateau, China

**Rui Zhang[1], Hongjuan Zhang[1]\*, Yunpeng Xing[1], Lian Xue[2]**

**1** College of Water Conservancy and Hydropower Engineering, Gansu Agricultural University, Lanzhou, China, **2** Lanzhou Agro-technical Research and Popularization Center, Lanzhou, China

\* zhj94amor@163.com

**Data Availability Statement:** Data are contained within the article.

**Funding:** This research has been supported by the "Fuxi youth talent training program" of Gansu

## Abstract

Knowledge on the variation of yield and water use efficiency under different mulching methods is important for guiding rained maize production in the Loess Plateau area. In this study, eight different plastic film mulching methods was established to analyze the maize growth, soil water content and soil temperature changes of dryland maize, and increase yield and water use efficiency (WUE). The field experiment was conducted in 2019, and eight treatments were set up, including a traditional flat planting without mulching (CK), ridge-furrow with ridges mulching black plastic film and furrows mulching straw (HJ), ridge-furrow with ridges mulching black plastic film and furrows bare (HL), ridge-furrow with ridges mulching liquid plastic film and furrows mulching straw (YJ), ridge-furrow with ridges mulching liquid plastic film and furrows bare (YL), ridge-furrow with ridges mulching biodegradable plastic film and furrows mulching straw (SJ), ridge-furrow with ridges mulching biodegradable plastic film and furrows bare (SL) and ridge-furrow with ridges bare and furrows mulching straw (NJ). Furthermore, the AHP-TOPSIS was employed to evaluate the optimal mulching method for maize. The results showed that compared with CK and NJ treatment, the soil water content and soil storage were significantly changes with other treatments in the reproductive period of maize. Among the six mulching methods, maize yield in HJ, HL, YJ, YL, SJ, and SL treatments were 46.28%, 61.95%, 70.30%, 51.02%, 52.02% and 53.53% significantly greater than CK treatment. In addition, dryland maize WUE was 66.53% and 84.01% higher in the YJ and YL treatments with ridges mulching liquid plastic film than in the CK treatment, respectively. The optimal treatments of economic benefits were YL and HJ. Through AHP-TOPSIS comprehensive analysis, the optimal mulching methods were YL and HJ treatment. Current field trials indicate that YL treatment could serve as a promising option to improve dryland maize yield, WUE, and reducing environmental risks in the Loess Plateau of China.

Agricultural University (Gaufx-03Y10 to R.Z), the construction project of the "Innovative Team for Water Saving Irrigation and Water Resource Regulation in Arid Irrigation Areas" in the discipline of water conservancy engineering at Gansu Agricultural University (2023-38 to R.Z) and the "Gansu Provincial Water Conservancy Science Experimental Research and Technology Promotion Program project" (23GSLK047 to R.Z, 24GSLK060 to R.Z).

**Competing interests:** The authors declare no conflict of interest.

## 1. Introduction

China's per capita possession of water resources is small and unevenly distributed over time and space [1, 2]. The population of northern China accounts for 40% of the total population of the country, and the arable land area accounts for 51% of the country [3], but the total amount of water resources only accounts for 20% of the country, and the per capita amount of water resources is only 1/25 of the world's per capita water resources [4, 5]. According to statistics, the annual water shortage in China's agricultural production is 26 billion $m^3$ [6], and water resources have become a pivotal factor limiting the expansion of China's agricultural industry [7].

Mulching technology is mainly used to block solar radiation and reduce water evaporation by covering the field with mulch [8], so as to increase temperature and moisture retention [9]. At present, mulch is widely used in arid areas, and its significant moisture retention and warming effect can significantly increase maize yield by 28.3% to 87.5% [10]. However, with the long-term use of plastic mulch, it appears that the mulch cannot be degraded quickly in the soil, and the mulch forms a residue in the farmland, resulting in the deterioration of the ecological environment of the farmland [11, 12]. In the context of green and sustainable agricultural development, new mulching methods such as liquid plastic film, biodegradable plastic film and straw return to the field are also attracting attention [13–16]. The return of mulching straw to the field has the advantages of inhibiting soil moisture evaporation, lowering soil temperature, and increasing soil fertility [17]. The application of mulching straw mulch has been demonstrated to effectively reduce soil temperature and increase in yield of maize in the arid area of the Loess Plateau [18]. The new mulching plastic film has also been concurrently studied extensively for maize yield improvement, improving yield and water use efficiency of maize in the Loess Plateau [19].

Soil water content and temperature are significant environmental factors that impact the growth and yield of crops [20, 21]. The Loess Plateau is located in the semi-arid region of inland China, where rainfall is low and evapotranspiration is high [22], leading to a scarcity of water available for agricultural production, which severely limits agricultural production and the lower economic benefits in the region [23, 24]. As a result, it is critical to understand how to effectively use water resources in order to develop water-saving agriculture. Maize is an important food crop in the Loess Plateau's semi-arid zone, and the study of water use efficiency of maize crops provides a theoretical basis for improving maize yield and economic benefits [25, 26]. In fact, improving water use efficiency, maize yield and economic benefits should be a comprehensive concept involving multiple factors. Whereas, white and black mulching raises environmental costs. Mulching improves economic efficiency, but increases environmental costs in Loess Plateau [27]. Therefore, optimizing mulch management by taking multiple factors into account to ensure maize yield and economic benefits lowest environmental costs is particularly important in Loess Plateau.

More studies have developed analytical models for evaluating multiple indicators using advanced methods [28]. In this study, a total of eight mulching methods were designed, and the changes in maize growth, yield, soil water content, soil temperature and economic benefits under different mulching methods were investigated. The main objectives of this study were: (1) to analyze the changes in maize plant height, leaf area, stem thickness, and dry matter under different ridge-furrow mulching methods; (2) to study the effects of different ridge and furrow mulched on soil water content, soil temperature, maize yield, water use efficiency and economic benefit (EB); and (3) exploring more effective dryland maize mulching methods based on AHP-TOPSIS model.

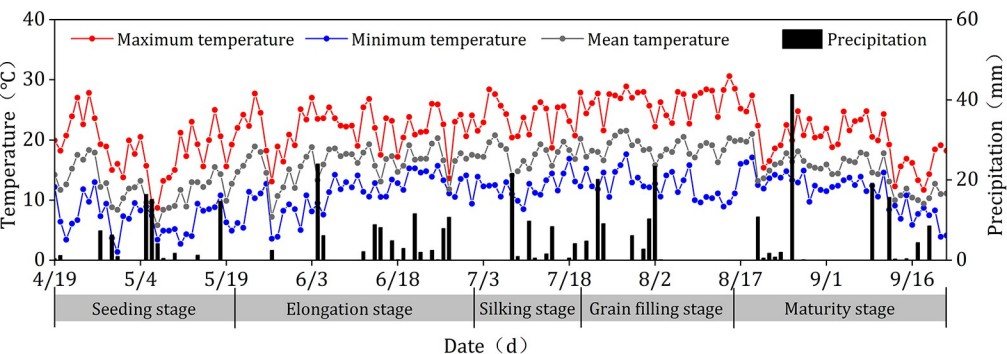

**Fig 1. Daily precipitation amounts and temperature during dryland maize growing seasons in 2019 at Dingxi, China.**

## 2. Materials and methods

### 2.1. Field experiment description

The field experiment was conducted from April to September 2019 at field trial Institute of Water Conservancy Science in Dingxi, Gansu Province, China (34°26′N, 103°52′E, elevation 1958m). The region is located in the semi-humid-mid temperate to semi-arid zone monsoon climate zone. The average annual sunshine hours in the region are 2114 to 2443 h, the average annual temperature is 5.7 to 7.7°C, the multi-year average precipitation is 400 mm (Fig 1), and the average annual evaporation is 1400 mm. The soil texture was a loam in the 0–20 cm soil layer, the soil color was yellow, and the soil water holding capacity was 23.7%. The nutrient in the soil property was pH 8.71, alkaline dissolved nitrogen (ADN) 32.25 mg kg$^{-1}$, total nitrogen (TN) 0.86 g kg$^{-1}$, nitrogen (N) 240.62 mg kg$^{-1}$, ammonium nitrogen (AN) 14.57 mg kg$^{-1}$, effective phosphorus (EP) 19.56 mg kg$^{-1}$, and fast-acting potassium (FP) 186.17 mg kg$^{-1}$.

### 2.2. Experimental design and field management

The maize (*Zea mays L.*) at a seeding density of 45000 plants ha$^{-1}$. The maize was planted using the ridge-furrow planting technique, where the width of the ridges was 60 cm, the height of the ridges was 25 cm and the width of the furrows was 60 cm. The eight mulching methods used in this experiment were: (1) ridge-furrow with ridges mulching black plastic film and furrows mulching straw (HJ), (2) ridge-furrow with ridges mulching black plastic film and furrows bare (HL), (3) ridge-furrow with ridges mulching liquid plastic film and furrows mulching straw (YJ), (4) ridge-furrow with ridges mulching liquid plastic film and furrows bare (YL), (5) ridge-furrow with ridges mulching biodegradable plastic film and furrows mulching straw (SJ), (6) ridge-furrow with ridges mulching biodegradable plastic film and furrows bare (SL), (7) ridge-furrow with ridges mulching bare and furrows mulching straw (NJ), and(8) a traditional flat planting without film mulching (CK) (Fig 2). In this study, biomass straw was used for furrows mulching straw. The mulching liquid plastic film is applied to the soil surface after preparation, which could facilitate the formation of a granular structure in the soil and indicate the formation of a layer of film in the soil. The liquid plastic film would gradually decompose into various nutrients required for crops after 60 days of film formation, so the liquid film is more environmentally protection and more convenient to use.

The width of mulching black plastic film, biodegradable plastic film, and liquid plastic film on ridge was 0.8 m, 0.9 m, and 1.0 m, respectively. Dryland maize was planted on April 19 and harvested on September 22 in 2019. Bottom fertilizer compound fertilizer 830 kg ha$^{-1}$

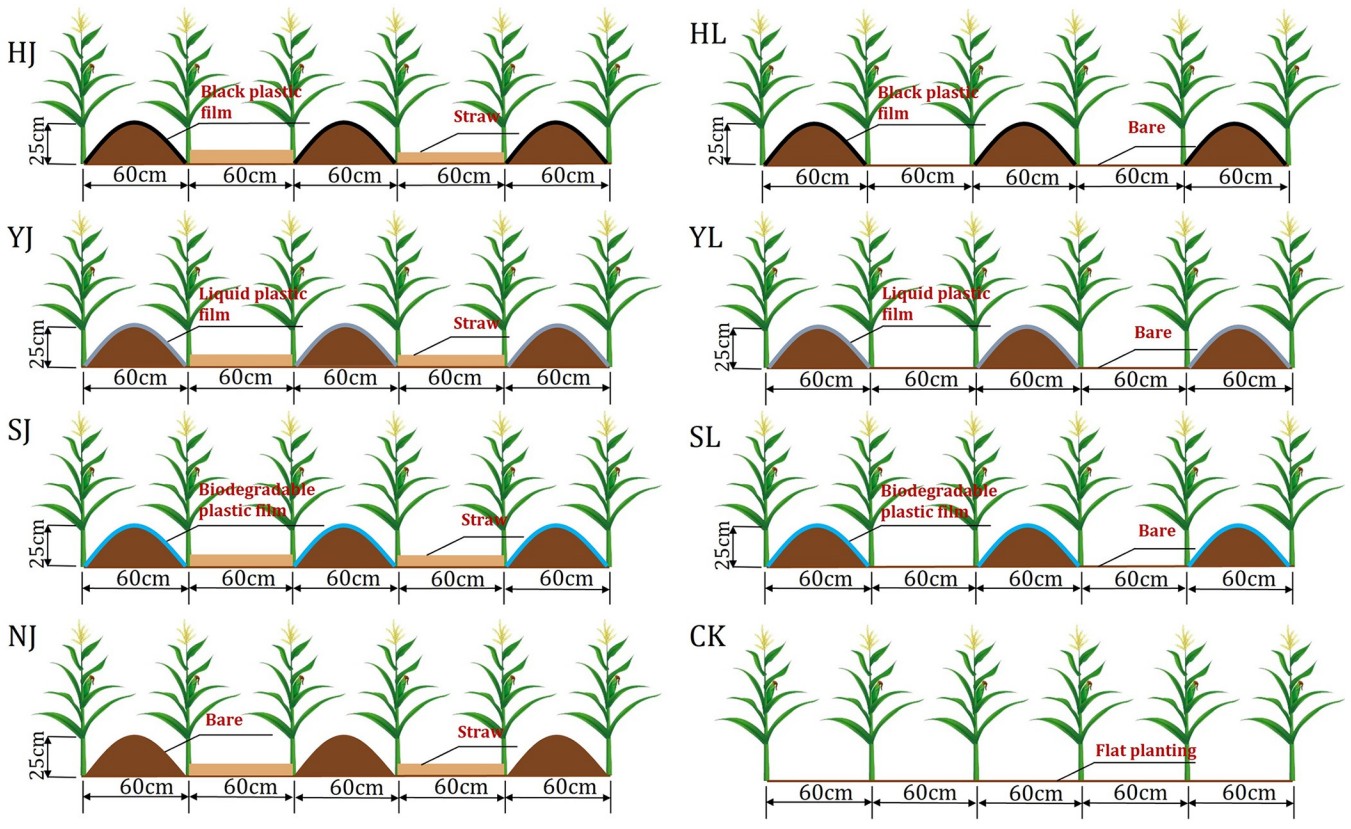

**Fig 2. A Schematic layout of the field experiment site.**

(N-P$_2$O$_5$-K$_2$O$_5$ = 18-12-10) and organic fertilizer 7500 kg ha$^{-1}$ were applied before maize sowing, and no topdressing was performed during the growth period. The experimental field was subject to regular weed control measures, and the field management practices employed were consistent with those typically observed in the local area during the maize growth period. The growth stages of maize were from April 19 to May 22 at seedling stage, from May 23 to June 28 at elongation stage, from June 29 to July 22 at silking stage, from July 23 to August 14 at grain filling stage, and from August 15 to September 22 at maturity stage in 2019.

## 2.3. Sampling and measurement

**2.3.1. Soil hydrothermal conditions.** Soil water content was measured with 10 cm, 20 cm, 40 cm, 60 cm, 80 cm and 100 cm soil layers at different fertility stages of dryland maize. Soil water content was measured every 15 days after maize planting, with additional observations before and after rainfall. Soil water content samples were collected at 10 cm intervals between two plants in the same row in each plot row using a 70 mm diameter portable auger. Soil water content was measured by drying method, each soil sample was measured three times, each treatment was repeated three times, and the average value was taken during each reproductive period. The soil water content (SWC) was calculated using the following equation:

$$SWC = (M1 - W2)/M2 \times 100\%$$ (1)

Where *SWC* is soil water content (%); *M1* is the measurement of wet weight (g); and *M2* is the

measurement of dry weight (g).

$$SWS = \sum_{i=1}^{n}(SWC_i H_i \times 10) \tag{2}$$

Where $SWS$ is soil water storage (mm); $SWC_i$ is soil water content in each different soil layer (%); $H_i$ is soil depth (cm); and $n$ is the number of soil layers.

Soil temperature was measured using a curved tube cryostat at depths of 5 cm, 10 cm, 15 cm, 20 cm and 25 cm. Measurements were taken once per reproductive period for three consecutive days, with daily observation times at 8:00 h, 10:00 h, 12:00 h, 14:00 h, 16:00 h, 18:00 h, and 20:00 h.

**2.3.2. Crop growth.** The height of maize plants was determined once in each reproductive period during the growth and development of maize. Five representative plants were selected in each plot, and then observed regularly. Before tasseling, the distance from the bottom of the maize stem to the furthest leaf part was determined, and after tasseling, the distance from the bottom of the stem to the top of the ear was determined.

Maize stem thickness was measured at the same time period as plant height, and the first node at the bottom of the stem was determined with a vernier caliper, and the longest and shortest parts were averaged.

Dry matter was measured on the ground during the growth and development of maize at each reproductive period. In the determination, three representative plants were selected, cut off from the bottom of plant stems, weighed the fresh weight of stems, leaves, and fruits were put into the drying oven first at 110°C for 30min, and dried at 85°C. The final weight after drying was measured.

Maize leaf area was measured in the same way as plant height, and the following equations were calculated:

$$S = \alpha \times L \times B \tag{3}$$

Where $\alpha$ is the maize leaf area coefficient, $\alpha = 0.75$ [29], $L$ is the leaf length (cm), $B$ is the leaf width (cm), and $S$ is the leaf area (cm$^2$).

**2.3.3. Grain yield.** Six plants in each plot were manually harvested at the full maturity stage of the maize and the yield were standardized to 14% moisture. The ear dry weight, ear length, ear diameter, number of ears per row, number of grains per row and 1000-kernels seed of harvested maize were measured and recorded respectively.

**2.3.4. Evapotranspiration (ET) and water use efficiency (WUE).** The evapotranspiration ($ET$) during the growth period was determined by using the water balance formula:

$$ET = P + W1 - W2 + I - R \tag{4}$$

where $P$ is the precipitation(mm); $I$ is the irrigation (mm), no irrigation in this experiment, $I = 0$ mm; $ET$ is the evapotranspiration (mm); $W1$ is water storage in the 0–100 cm soil layer in preplant (mm); $W2$ is water storage in the 0–100 cm soil layer in post-harvest (mm), and $R$ is surface runoff (mm).

The water use efficiency ($WUE$, kg ha$^{-1}$ mm$^{-1}$) was calculated using the following formula in this study [30]:

$$WUE = \frac{Y}{ET} \tag{5}$$

Where $Y$ is the yield (kg ha$^{-1}$) and $ET$ is the total evapotranspiration over the growth period (mm).

**2.3.5. Economic benefit (EB).** The economic benefit (*EB*) was calculated using the following equation:

$$EB = P - TC \tag{6}$$

Where *P* is profit from maize yield (CNY ha$^{-1}$), and *TC* is the total cost, in this study, there is no irrigation for dryland planting, so the total cost is the material cost.

## 2.4. Comprehensive evaluation

### 2.4.1. Analytical Hierarchy Process (AHP).
The Analytical Hierarchy Process (AHP) method is a valuable tool for decision analysis, as it enables the evaluation of complex systems comprising numerous interrelated and mutually restricted factors from diverse perspectives. The problem is hierarchised and structured according to the decision-making objectives of the system, creating a hierarchical structure and forming a multilevel analytical structural model. The multilevel structure is generally divided into objective, criterion and indicator levels, with the relative importance of the lower level to the upper level used to evaluate the weights of the factors [31]. The AHP needs to analyze the indicators hierarchically. This study determines the target layer, the criterion layer and the index layer, and establishes the evaluation index system (Fig 3).

1. *Constructing a judgment matrix.* The judgment matrix is the basic information of the analytic hierarchy process, and it is also an important basis for the relative comparison calculation structure. The equation of judgment matrix *A* is as follows:

$$A = \begin{bmatrix} a_{11} & a_{12} & \cdots & a_{1n} \\ a_{21} & a_{22} & \cdots & a_{2n} \\ \vdots & \vdots & \ddots & \vdots \\ a_{n1} & a_{n2} & \cdots & a_{nn} \end{bmatrix} \tag{7}$$

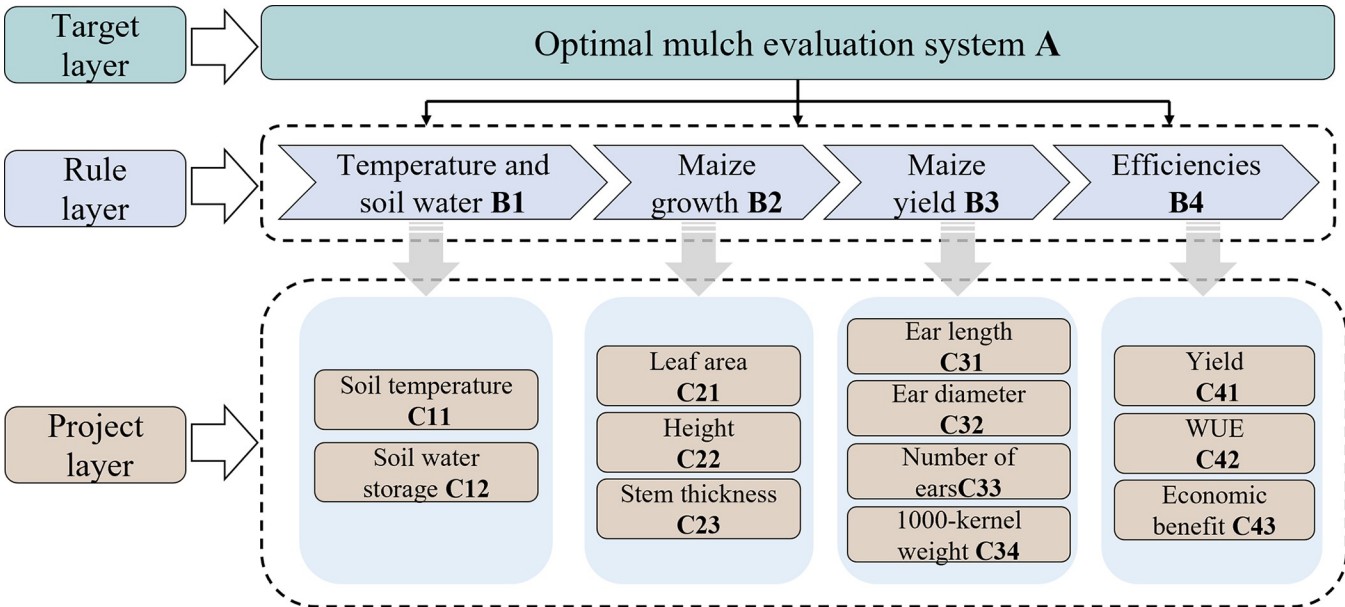

**Fig 3. Hierarchy structure of Analytical Hierarchy Process (AHP).**

Where $a_{ij}$ ($i, j$ = 1, 2, 3. . ., n) are represents the importance value of the $i$-th factor relative to the $j$-th factor.

2. *Calculation weight.* The normalized processing formula is:

$$b_{ij} = a_{ij} / \sum_{i=1}^{n} a_{ij} (i, j = 1, 2, 3 \ldots n) \tag{8}$$

Where $a_{ij}$ is the element of the $i$-th row and $j$-th column of the judgment matrix $A$, and $b_{ij}$ is the element after normalization.
The weight vector formula is:

$$\tilde{w}_i = \sum_{j=1}^{n} b_{ij} \tag{9}$$

$$w_i = \tilde{w}_i / \sum_{i=1}^{n} \tilde{w}_i \tag{10}$$

$$\lambda_{\max} = \sum_{i=1}^{n} \frac{(AW)_i}{nw_i} \tag{11}$$

Where $\lambda_{\max}$ is eigenvalue of maximum, $A$ is judgment matrix, and $W$ is eigenvector.

3. *Consistency test.* The higher complexity of the indicator system and the higher subjectivity of the hierarchical analysis method, the judgment matrix may have errors, in order to ensure the reasonableness of the weight vector, so the consistency test of the judgment matrix. In this study, the consistency test was calculated using $CR$:

$$CR = \frac{CI}{RI} = \frac{\lambda_{max} - n}{(n - 1)RI} \tag{12}$$

Where $n$ is the order of the judgment matrix, $CI$ is the consistency test index, and $RI$ is the average stochastic consistency index, which is related to the order n, and the value of the criterion was shown in Table 1. If the judgement matrix passes the consistency test, it can be concluded that the construction of the multilevel judgement matrix is in accordance with the mathematical logic, and that the weight vector can be calculated based on this matrix.

### 2.4.2. TOPSIS comprehensive evaluation method.

*(1) Evaluation matrix normalization.* The normalization equation of matrix R is as follows:

$$r_{ij} = \frac{u_{ij}}{\sqrt{\sum_{j=1}^{m} u_{ij}^2}} \quad j = 1, 2, \ldots, m \; i = 1, 2, \ldots, n \tag{13}$$

Where $r_{ij}$ is the normalized evaluation matrix element, and $u_{ij}$ is the value of the $i$-th index of the $j$-th scheme.

**Table 1. Average random consistency index value.**

| Order n | 1 | 2 | 3 | 4 | 5 | 6 | 7 | 8 | 9 |
|---------|------|------|------|------|------|------|------|------|------|
| RI | 0.00 | 0.00 | 0.58 | 0.90 | 1.12 | 1.24 | 1.32 | 1.41 | 1.45 |

*(2) Constructing weighted normalized decision matrix.* The formula of constructing weighted normalized decision matrix is as follows:

$$v_{ij} = W_i \times r_{ij} \quad j = 1, 2, \ldots, n \tag{14}$$

Where $v_{ij}$ is the element of weighted normalized decision matrix element.

*(3) Determining positive and negative ideal solutions.* A positive ideal solution $S^+$ denotes the optimal ideal solution, while a negative ideal solution $S^-$ denotes the least ideal solution.

$$S^+ = \{v_1^+, \ldots, v_i^+\} = \{(max_j v_{ij} | i \in I'), ((min_j v_{ij} | i \in I'')\} \tag{15}$$

$$S^- = \{v_1^+, \ldots, v_i^+\} = \{(min_j v_{ij} | i \in I'), ((max_j v_{ij} | i \in I'')\} \tag{16}$$

Where $I'$ is the larger the value index set, and $I''$ is the smaller the better the index set.

*(4) Calculate the separation measure.* The positive and negative ideal solution distance calculation formula is as follows:

$$D_j^+ = \sqrt{\sum_{i=1}^{n} (v_{ij} - v_i^+)^2} \quad j = 1, 2, \ldots, m \tag{17}$$

$$D_j^- = \sqrt{\sum_{i=1}^{n} (v_{ij} - v_i^-)^2} \quad j = 1, 2, \ldots, m \tag{18}$$

Where $D_j^+$ is the distance of positive ideal solution, and $D_j^-$ is the distance of negative ideal solution.

*(5) Priority ordering.* The relative closeness formula is as follows:

$$C = \frac{D_j^-}{D_j^+ + D_j^-} \quad j = 1, 2, \ldots, m \tag{19}$$

Where $C$ is the relative closeness of the program to the ideal solution, which is generally in the range of (0,1), the closer to 1 means the closer to the positive ideal solution, and the closer to 0 means the closer to the negative ideal solution [32].

**2.4.3. AHP-TOPSIS integrated judgement model.** The evaluation matrix is constructed through a process of closeness analysis utilising the TOPSIS method. The combined weight calculated by the AHP method is incorporated into the vector Q formula, which represents the comprehensive evaluation result of the evaluation is as follows:

$$Q = W \times C \tag{20}$$

Where W is the weight of criterion layer calculated by AHP method, and $C$ is the relative closeness of the program to the ideal solution.

## 2.5. Statistical analysis

All data collected are expressed as the mean value of three replicates. Crop growth data processing was conducted in Microsoft Excel 2016. Data were analyzed by analysis of variance through SPSS 20.0 and significant differences were analyzed. Multiple comparisons were performed using the least significant difference (LSD) tests at P < 0.05. Figures were created through Origin 2022.

## 3. Results

### 3.1. Soil water content and soil water storage

The soil water content was influenced by mulching method and precipitation, and varied with the growth period of dryland maize (Fig 4). Among the six mulching treatments, soil water content value was generally higher before silking stage but decreased significantly after silking stage. Soil water content of different treatments of maize showed greater performance at seedling and elongation stage, with soil water content values greater than 17.00% and were 17.08–19.93%. In the seedling and elongation stage of maize, the precipitation was relatively concentrated and frequent, and the precipitation was 145.51 mm, accounting for 37.26% of the precipitation in the growth period, resulting in a large soil water content value of maize in this stage. The soil water content of different treatments was lower than 15.50% during the grain filling stage of maize. In the grain filling stage, maize was in the period of vigorous growth, crop transpiration was in the most vigorous stage of the whole productive period, and the temperature in this stage was also higher, resulting in lower soil water content value.

In different growth stages of maize, the soil water content of different treatments showed different changes in 0–100 cm soil layer (Fig 5). The fluctuation of soil water content in HJ, HL, NJ and CK treatments were greater than that in SJ, SL, YJ, and YL treatments at elongation and silking stages of maize (P<0.05). At the elongation stage of maize, the soil water content values under the 0–100 cm soil layer of YJ, YL, SJ and SL treatments were 17.30–20.01%, 16.67–19.62%, 18.49–20.77% and 18.32–20.79%, respectively. At the silking stage of maize, the difference between the maximum and minimum values of soil water content in different soil layers of 0–100 cm under YJ, YL, SJ and SL treatments were 3.59%, 4.19%, 4.58% and 3.92%, respectively. However, in the growth stage of maize, compared with YJ, YL, SJ and SL, the soil water content of HJ, HL, NJ and CK treatments showed large fluctuations. The soil water content of each treatment under the 0–60 cm soil layer as a whole showed that the elongation stage > silking stage > grain filling stage. The difference of soil water content in 0–100 cm soil layer under each treatment was small when maize reached maturity stage, which was mainly due to the gradual maturity of crops.

Soil water storage in the 0–100 cm soil layer was significantly different between post-harvest and pre-planting under different mulching treatments (P<0.05) (Fig 6). The SJ, SL, YJ, YL, HJ,

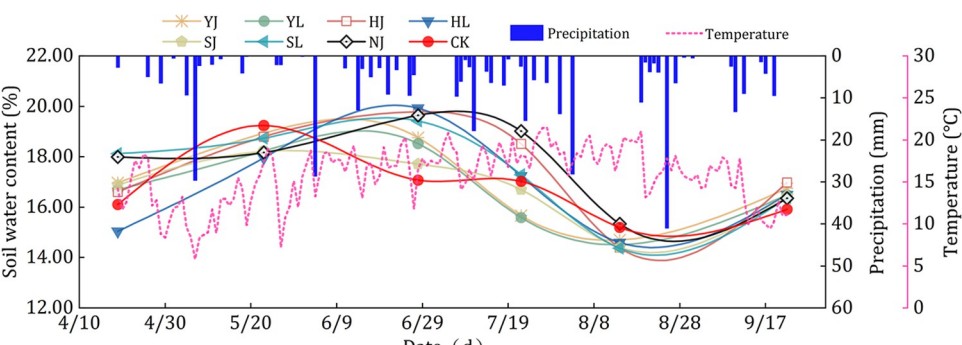

**Fig 4. Patterns of soil water content changes during maize fertility under different treatments.** Note: HJ is ridge-furrow with ridges mulching black plastic film and furrows mulching straw; HL is ridge-furrow with ridges mulching black plastic film and furrows bare; YJ is ridge-furrow with ridges mulching liquid plastic film and furrows mulching straw; YL is ridge-furrow with ridges mulching liquid plastic film and furrows bare; SJ is ridge-furrow with ridges mulching biodegradable plastic film and furrows mulching straw; SL is ridge-furrow with ridges mulching biodegradable plastic film and furrows bare; NJ is ridge-furrow with ridges mulching bare and furrows mulching straw; and CK is a traditional flat planting without film mulching.

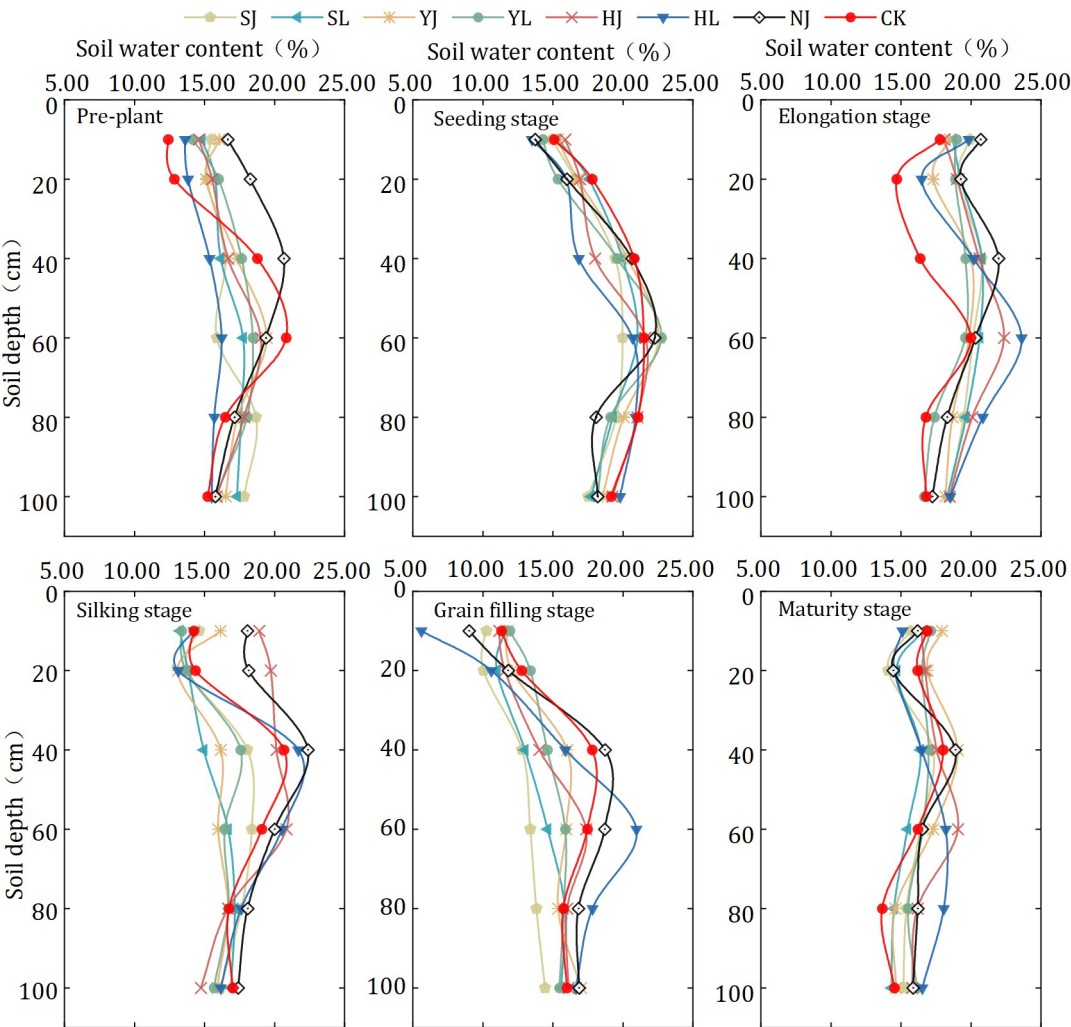

**Fig 5. Dynamic change of soil water content in the 0–100 cm soil layer among different mulching treatments at the stages of maize growth.** Note: HJ is ridge-furrow with ridges mulching black plastic film and furrows mulching straw; HL is ridge-furrow with ridges mulching black plastic film and furrows bare; YJ is ridge-furrow with ridges mulching liquid plastic film and furrows mulching straw; YL is ridge-furrow with ridges mulching liquid plastic film and furrows bare; SJ is ridge-furrow with ridges mulching biodegradable plastic film and furrows mulching straw; SL is ridge-furrow with ridges mulching biodegradable plastic film and furrows bare; NJ is ridge-furrow with ridges mulching bare and furrows mulching straw; and CK is a traditional flat planting without film mulching.

HL and NJ treatments were not significantly different from the CK treatment prior to maize planting (P>0.05). Meanwhile, the difference was -0.65%, -0.46%, 2.23%, 0.79%, 0.15%, 1.22% and 3.42% compared to CK treatment. The YJ, YL, HJ, HL and NJ treatments were significantly higher than the CK treatment by 11.01%, 7.75%, 16.88%, 13.47% and 10.11%, respectively, in the post-harvest. Whereas, SJ and SL treatments had relatively small differences compared to CK treatment. Overall, the soil water storage capacity of maize grown under furrow cover was higher than that of conventional flat crop.

## 3.2. Soil temperature

Different mulching treatments showed varying effects on soil temperature during the different growth period. In this study, the same weather during the growth period were selected to

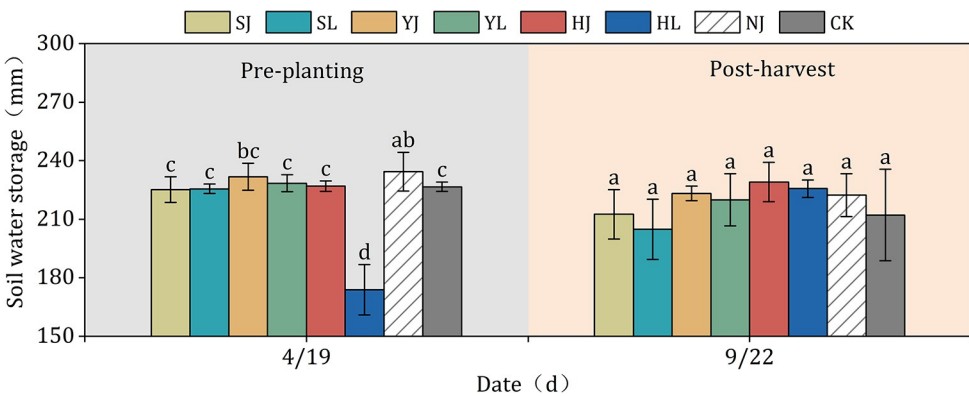

**Fig 6. Dynamic change of soil water storage in the 0–100 cm layer among various treatments at the pre-planting and post-harvest.** Different lowercase letters above bars indicate significant differences (P < 0.05) according to LSD. Note: HJ is ridge-furrow with ridges mulching black plastic film and furrows mulching straw; HL is ridge-furrow with ridges mulching black plastic film and furrows bare; YJ is ridge-furrow with ridges mulching liquid plastic film and furrows mulching straw; YL is ridge-furrow with ridges mulching liquid plastic film and furrows bare; SJ is ridge-furrow with ridges mulching biodegradable plastic film and furrows mulching straw; SL is ridge-furrow with ridges mulching biodegradable plastic film and furrows bare; NJ is ridge-furrow with ridges mulching bare and furrows mulching straw; and CK is a traditional flat planting without film mulching.

compare the effects of different mulching measures on soil temperature. The diurnal variation of soil temperature in different growth stages was similar (Fig 7). However, compared to CK and NJ treatment, the mulching treatments, especially SJ, SL, YJ and YL significantly decreased soil temperature from elongation stage to grain filling stage (P<0.05). At the elongation stage, compared to CK treatment, the maximum and minimum differences in daily soil temperature variations were 3.04˚C (P<0.05), 3.48˚C (P<0.05), 3.66˚C (P<0.05) and 2.44˚C (P<0.05) for SJ, SL, YJ and YL treatments, respectively. At the silking stage, compared to CK, the maximum and minimum differences in daily soil temperature variations were 3.04˚C (P<0.05), 3.64˚C (P<0.05), 2.98˚C (P<0.05) and 1.36˚C (P<0.05) for SJ, SL, YJ and YL, respectively. The daily variation in soil temperature was greater in the CK and NJ treatments than in SJ, SL, YJ, YL, HJ and HL from the elongation stage to the grain filling stage (P>0.05).

### 3.3. Height, leaf area and stem thickness

Compared to CK and NJ treatments, mulching treatments significantly increased height, leaf area and stem thickness in 2019 (Fig 8). Relative to CK and NJ treatments, the average height of the YJ, YL, HJ, HL, SJ and SL treatments were improved by 39.97%, 39.46%, 40.69%, 38.47%, 34.59% and 32.73% (P<0.05), and 37.16%, 36.66%, 37.86%, 35.69%, 31.89% and 30.06% (P<0.05) at the maize growth period, respectively. The average plant height of the YJ, YL, HJ and HL four treatments were higher, with an average value of 220.56–224.09 cm (P<0.05) at the growth period. In general, the height of all mulching treatments was higher than that of CK and NJ treatments.

Compared to CK and NJ treatments, leaf area in CK and NJ were significantly lower than that in YJ, YL, HJ, HL, SJ and SL treatments after the seeding stage in 2019 (Fig 8). Among the four mulching treatments, leaf area in CK and NJ treatments less than 21.41–54.27% (P<0.05) at the growth period. In addition, the leaf area average showed HL (7357.30 cm$^2$) > HJ (6824.42 cm$^2$) > YJ (6113.73 cm$^2$) > YL (6088.03 cm$^2$) > SL (6061.23 cm$^2$) > SJ (5965.64 cm$^2$) at the silking stage, grain filling stage and the maturity stage (P<0.05).

Mean values of stem thickness of YJ, YL, HJ, HL, SJ, and SL treatments were significantly higher than those of CK treatments during the complete reproductive period of maize

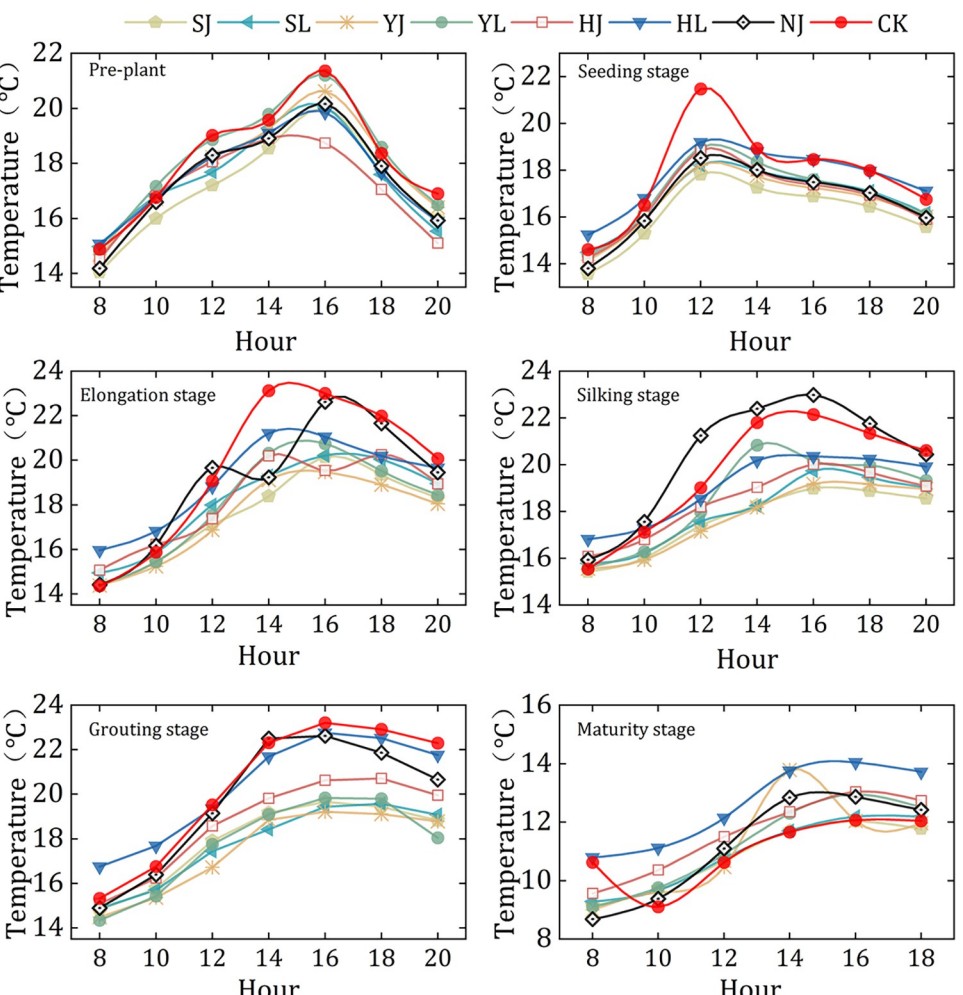

**Fig 7. Dynamic change of daily soil temperature in the 0–25 cm soil layer among different treatments at the stages of maize fertility.** Note: HJ is ridge-furrow with ridges mulching black plastic film and furrows mulching straw; HL is ridge-furrow with ridges mulching black plastic film and furrows bare; YJ is ridge-furrow with ridges mulching liquid plastic film and furrows mulching straw; YL is ridge-furrow with ridges mulching liquid plastic film and furrows bare; SJ is ridge-furrow with ridges mulching biodegradable plastic film and furrows mulching straw; SL is ridge-furrow with ridges mulching biodegradable plastic film and furrows bare; NJ is ridge-furrow with ridges mulching bare and furrows mulching straw; and CK is a traditional flat planting without film mulching.

($P<0.05$) (Fig 8). The mean values of maize stem thickness were higher in YJ, YL, HJ, HL, SJ and SL treatments than in CK and NJ treatments by 16.37% ($P<0.05$), 14.59% ($P<0.05$), 22.02% ($P<0.05$), 16.80% ($P<0.05$), 17.62% ($P<0.05$) and 18.45% ($P<0.05$), and 19.90% ($P<0.05$), 18.06% ($P<0.05$), 25.72% ($P<0.05$), 20.34% ($P<0.05$), 21.18% ($P<0.05$) and 22.04% ($P<0.05$), respectively. At elongation stage, the difference of maize stem thickness diameter between YJ, YL, HJ, HL, SJ, and SL treatments and CK treatment was the largest, which was 45.86%, 47.99%, 49.43%, 50.29%, 53.30%, and 48.56%, respectively.

## 3.4. Aboveground dry matter

Dry matter per plant was significantly greater for the YJ, YL, HJ, HL, SJ and SL treatments than for the CK treatment after the seeding stage ($P<0.05$) (Fig 9). In the elongation stage and silking stage, the dry matter quality of HJ and HL treatments was significantly higher than that

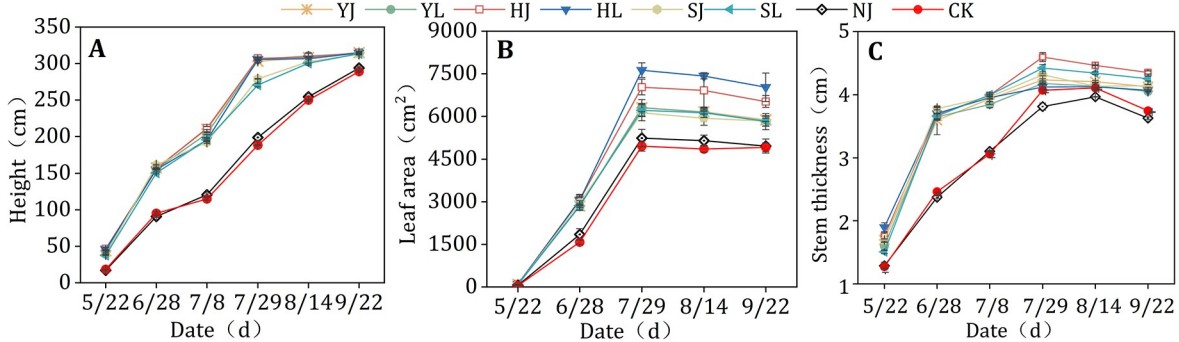

**Fig 8. Effects of different mulching treatments on the height, leaf area and stem thickness of rained maize in 2019.** Error bars indicate standard deviation (P<0.05). Note: HJ is ridge-furrow with ridges mulching black plastic film and furrows mulching straw; HL is ridge-furrow with ridges mulching black plastic film and furrows bare; YJ is ridge-furrow with ridges mulching liquid plastic film and furrows mulching straw; YL is ridge-furrow with ridges mulching liquid plastic film and furrows bare; SJ is ridge-furrow with ridges mulching biodegradable plastic film and furrows mulching straw; SL is ridge-furrow with ridges mulching biodegradable plastic film and furrows bare; NJ is ridge-furrow with ridges mulching bare and furrows mulching straw; and CK is a traditional flat planting without film mulching.

of other treatments. In the grain filling stage and maturity stage, there was no significant difference in dry matter quality between YJ, YL, HJ, HL, SJ and SL treatments (P>0.05). Compared to CK and NJ treatments, the average dry matter of the YJ, YL, HJ, HL, SJ and SL treatments were increased by 58.39%, 62.20%, 76.15%, 74.04%, 57.39% and 57.43% (P<0.05), and 12.98%, 15.69%, 25.64%, 24.14%, 12.26% and 12.29% (P<0.05) at the maize growth period, respectively.

## 3.5. Yield and yield components

Table 2 maize yield and yield components of maize under different mulching treatments. The YJ, YL, HJ, HL, SJ and SL treatments were higher to the CK treatment of ear diameter, number of ears, 1000-kernel weight, and yield. All treatments were not significantly of ear length, ear diameter, number of ears (P>0.05). Ridge-furrow mulching planting significantly increased maize yield. Compared to CK and NJ treatments, YJ, YL, HJ, HL, SJ and SL treatments

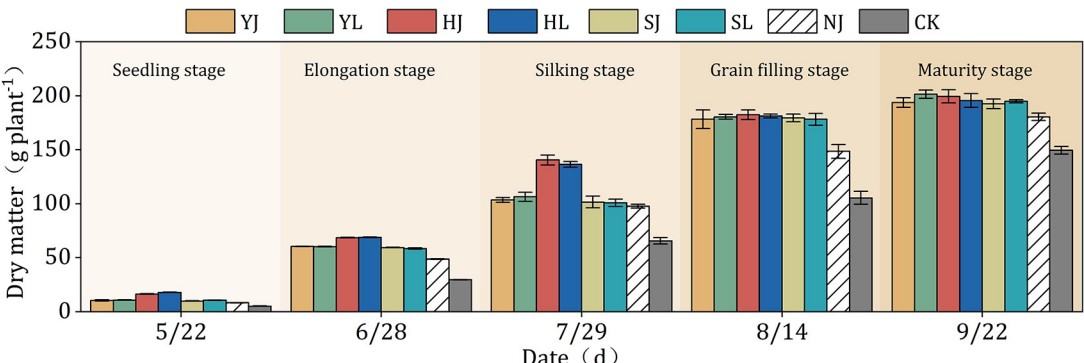

**Fig 9. Effect of different mulching treatments on the dry matter of dryland maize in 2019.** Error bars indicate standard deviation (P<0.05). Note: HJ is ridge-furrow with ridges mulching black plastic film and furrows mulching straw; HL is ridge-furrow with ridges mulching black plastic film and furrows bare; YJ is ridge-furrow with ridges mulching liquid plastic film and furrows mulching straw; YL is ridge-furrow with ridges mulching liquid plastic film and furrows bare; SJ is ridge-furrow with ridges mulching biodegradable plastic film and furrows mulching straw; SL is ridge-furrow with ridges mulching biodegradable plastic film and furrows bare; NJ is ridge-furrow with ridges mulching bare and furrows mulching straw; and CK is a traditional flat planting without film mulching.

**Table 2. Comparison of maize yield and yield components in different mulching treatments.**

| Treatment | Ear length (cm) | Ear diameter (cm) | Number of ears (row1) | 1000-kernel weight (g) | Yield (kg ha$^{-1}$) |
|---|---|---|---|---|---|
| YJ | 25.51±1.26a | 5.37±0.11a | 14.56±0.16ab | 428.67±16.44a | 8257.03±437.33b |
| YL | 25.42±1.77a | 5.43±0.07a | 14.56±0.16ab | 423.81±9.66a | 9141.75±1227.94ab |
| HJ | 26.14±0.51a | 5.40±0.05a | 15.17±0.14a | 426.30±2.01a | 9613.50±1380.44a |
| HL | 25.76±1.05a | 5.31±0.27a | 14.56±0.87ab | 415.88±19.70a | 8525.00±1411.42ab |
| SJ | 26.97±0.57a | 5.56±0.04a | 14.67±0.27ab | 435.21±15.18a | 8581.25±356.49ab |
| SL | 26.10±1.52a | 5.52±0.11a | 14.67±0.54ab | 401.45±24.38a | 8666.37±1245.54ab |
| NJ | 26.53±0.27a | 5.27±0.36a | 14.89±0.42a | 334.95±41.99b | 5112.88±228.41c |
| CK | 25.58±0.47a | 5.36±0.03a | 13.64±0.27b | 329.21±32.39b | 5644.77±565.37c |

Note: Different lowercase letters indicate significant differences between treatments for maize growth components (P < 0.05). HJ: ridge-furrow with ridges mulching black plastic film and furrows mulching straw, HL: ridge-furrow with ridges mulching black plastic film and furrows bare, YJ: ridge-furrow with ridges mulching liquid plastic film and furrows mulching straw, YL: ridge-furrow with ridges mulching liquid plastic film and furrows bare, SJ: ridge-furrow with ridges mulching biodegradable plastic film and furrows mulching straw, SL: ridge-furrow with ridges mulching biodegradable plastic film and furrows bare, NJ: ridge-furrow with ridges mulching bare and furrows mulching straw, and CK: a traditional flat planting without film mulching.

significantly increased yield by 46.28%, 61.95%, 70.30%, 51.02%, 52.02% and 53.53% (P<0.05), and 61.49%, 78.80%, 88.03%, 66.74%, 67.84% and 69.50% (P<0.05), respectively. The HJ treatment had the highest value of yield, and the NJ treatment had the lowest value of yield. The HJ and SJ treatment of 1000-grain weight was significantly higher than CK treatment, which was higher by 29.49% and 32.20%, respectively (P<0.05).

## 3.6. Water use efficiency (WUE)and economic benefits

The maize water use efficiency (WUE) and economic benefits of maize under different mulching treatments (Tables 3 and 4). The maize WUE had significantly in different mulching treatments (P<0.05). The maize evapotranspiration of different mulching treatments was mainly SL > NJ > SJ > CK > YJ > YL > HJ > HL. Compared to CK treatment, YL, HJ and HL treatments increased yield by 1.44%, 1.45%, 4.11% and 18.78% (P<0.05), respectively. The WUE in the YJ, YL, HJ, HL, SJ and SL treatments increased significantly by 48.08%, 63.58%, 70.30%, 78.76%, 53.07% and 48.63%, and 66.53% (P < 0.05), 84.01%, 88.03%, 101.08%, 72.19% and

**Table 3. Comparison of water use efficiency in different mulching treatments.**

| Treatment | Yield (kg ha$^{-1}$) | ET (mm) | WUE (kg ha$^{-1}$ mm$^{-1}$) |
|---|---|---|---|
| YJ | 8257.03±437.33c | 412.94±4.05ab | 20.00±1.13d |
| YL | 9141.75±1227.94c | 412.93±11.29ab | 22.10±2.56cd |
| HJ | 9613.50±1380.44bc | 402.35±7.39b | 23.89±3.00bc |
| HL | 8525.00±1411.42c | 352.66±14.12c | 24.15±3.83bcd |
| SJ | 8581.25±356.49c | 415.08±2.91ab | 20.68±0.93d |
| SL | 8666.37±1245.54c | 430.84±6.09a | 20.08±2.62d |
| NJ | 5112.88±228.41d | 425.53±8.08ab | 12.01±0.31e |
| CK | 5644.77±565.37d | 418.90±21.89ab | 13.51±1.45e |

Note: Different lowercase letters indicate significant differences between treatments for maize growth components (P < 0.05). HJ: ridge-furrow with ridges mulching black plastic film and furrows mulching straw, HL: ridge-furrow with ridges mulching black plastic film and furrows bare, YJ: ridge-furrow with ridges mulching liquid plastic film and furrows mulching straw, YL: ridge-furrow with ridges mulching liquid plastic film and furrows bare, SJ: ridge-furrow with ridges mulching biodegradable plastic film and furrows mulching straw, SL: ridge-furrow with ridges mulching biodegradable plastic film and furrows bare, NJ: ridge-furrow with ridges mulching bare and furrows mulching straw, and CK: a traditional flat planting without film mulching.

**Table 4. Comparison of economic benefit in different mulching treatments.**

| Treatment | Basic input (CNY ha⁻¹) | Fertilizer input (CNY ha⁻¹) | Total input (CNY ha⁻¹) | Total income (CNY ha⁻¹) | Economic benefit (CNY ha⁻¹) |
|---|---|---|---|---|---|
| YJ | 1800.00 | 750.00 | 2550.00 | 11890.12 | 9340.12 |
| YL | 1800.00 | 750.00 | 2550.00 | 13164.12 | 10614.12 |
| HJ | 1980.00 | 750.00 | 2730.00 | 13843.44 | 11113.44 |
| HL | 1980.00 | 750.00 | 2730.00 | 12276.00 | 9546.00 |
| SJ | 4500.00 | 750.00 | 5250.00 | 12357.00 | 7107.00 |
| SL | 4500.00 | 750.00 | 5250.00 | 12479.57 | 7229.57 |
| NJ | 1500.00 | 750.00 | 2250.00 | 7362.55 | 5112.55 |
| CK | 1500.00 | 750.00 | 2250.00 | 8128.47 | 5878.47 |

Note: HJ: ridge-furrow with ridges mulching black plastic film and furrows mulching straw, HL: ridge-furrow with ridges mulching black plastic film and furrows bare, YJ: ridge-furrow with ridges mulching liquid plastic film and furrows mulching straw, YL: ridge-furrow with ridges mulching liquid plastic film and furrows bare, SJ: ridge-furrow with ridges mulching biodegradable plastic film and furrows mulching straw, SL: ridge-furrow with ridges mulching biodegradable plastic film and furrows bare, NJ: ridge-furrow with ridges mulching bare and furrows mulching straw, and CK: a traditional flat planting without film mulching.

67.19% ($P < 0.05$), respectively, relative to CK and NJ treatments. Compared to CK treatment, YJ, YL, HJ, HL, SJ and SL treatments increased economic benefit by 37.06%, 44.62%, 47.10%, 38.42%, 17.29% and 18.69%, respectively. Compared with CK treatment, better economic benefits were YL and HJ treatments, respectively.

## 3.7. AHP-TOPSIS method to evaluate optimal mulching treatment

Table 5 shows the metric values of the combined AHP-TOPSIS judgments for the different mulching treatments. Among them, the composite indicator Q of the HJ mulching treatment is the largest, with a maximum of 0.999, and followed by the YL and SL mulching treatments, which are 0.936 and 0.732, respectively. The mulching treatments with composite indicator Q greater than 0.8 and closer to 1 were the HJ and YL treatments. Therefore, the optimal mulching treatments were HJ and YL, respectively.

**Table 5. AHP-TOPSIS evaluation calculation result.**

| Treatment | Composite indicator Q | Result |
|---|---|---|
| YJ | 0.722 | 5 |
| YL | 0.936 | 2 |
| HJ | 0.999 | 1 |
| HL | 0.728 | 4 |
| SJ | 0.711 | 6 |
| SL | 0.732 | 3 |
| NJ | 0.001 | 8 |
| CK | 0.133 | 7 |

Note: HJ: ridge-furrow with ridges mulching black plastic film and furrows mulching straw, HL: ridge-furrow with ridges mulching black plastic film and furrows bare, YJ: ridge-furrow with ridges mulching liquid plastic film and furrows mulching straw, YL: ridge-furrow with ridges mulching liquid plastic film and furrows bare, SJ: ridge-furrow with ridges mulching biodegradable plastic film and furrows mulching straw, SL: ridge-furrow with ridges mulching biodegradable plastic film and furrows bare, NJ: ridge-furrow with ridges mulching bare and furrows mulching straw, and CK: a traditional flat planting without film mulching.

## 4. Discussion

In dryland areas, precipitation is often the only source of soil water content, so maintaining precipitation in the soil plays a critical role in crop growth and development. Compared without planting on mulching, ridge-furrow mulching has become a research hotspot for soil water content and temperature regulation of crop growth [33, 34]. Ridge-furrow mulching could increase precipitation in the furrow by collecting rainwater, and at the same time regulate the soil water and heat environment to meet the growth of crops, thus making full use of trace precipitation and ineffective precipitation in the field [35–37]. The ridge-furrow with plastic film and straw mulching served as physical barriers to prevent rainwater from running off or evaporating, thereby increased soil moisture infiltration and storage, so that there was no significant difference in post-harvest soil water storage obtained in this study, under the different mulching treatments [38]. In this study, the fluctuations of soil water content change under HJ, HL, YJ, YL, SJ, and SL treatments were smaller than those of NJ and CK treatments, so that the ridge-furrow mulching had a better effect of maintaining soil water content (Figs 3–5). Compared to traditional level ploughing, ridges plus mulch cover dramatically change the horizontal distribution and redistribution of soil water content (Fig 6).

The soil temperature has a significant effect on crop growth [39], especially in the cool, arid region of the Loess Plateau [40]. Average daily soil temperatures were less for mulching black plastic film (11.59–20.36˚C) than for biodegradable plastic film and liquid plastic film (10.88–18.59˚C) at maize growth period (Fig 7). This may be due to the mulching black plastic film could absorb more photosynthetically active radiation, which made the soil temperature under mulching black plastic film higher than that of mulching liquid and biodegradable plastic film [41].

Crop growth indicators such as plant height, leaf area and dry matter were key determinants of crop yield [42]. It was found that furrow mulching resulted in a significant increase in dry matter and leaf area of the crop [43]. Changes in maize stem thickness indicate the ability to provide nutrients, and the larger the stem thickness, the more resistant the maize is to downfall [44]. Dry matter accumulation was an indicator for visual analysis of maize growth, is a key indicator of the sustainability of dryland agriculture and favorable hydrothermal conditions resulting from mulching can significantly increase dry matter [45]. Numerous studies have been found that mulching treatments result in significant increases in plant height, leaf area, and dry matter of maize, mainly due to reduced evaporation of soil moisture from the mulched soil and the stabilisation of soil temperature in the appropriate range to promote plant growth [24]. Our study obtained similar results that the plant height, leaf area, stem diameter and dry matter mass of crops under mulching plastic film were significantly higher than those without mulching (CK and NJ) in the different growth stages of maize (Figs 8 and 9). This study showed that maize plant height, leaf area, stem thickness, and dry matter mass were significantly higher in the ridge-furrow mulching treatments than in the NJ and CK treatments without mulching.

Improving precipitation water use efficiency and yield are two of the major objectives of development for dryland agricultural, especially in the arid areas of the Loess Plateau. In semi-arid regions, mulching could significantly improve food yield and water use efficiency for a wide range of crops, including potato [16], wheat [46], and maize [47]. Plastic film mulching can reduce the interannual variation of maize yield and improve crop productivity, so it is an attractive choice for the development of dryland agriculture. In this study, furrow ground cover also significantly increased maize yield and water use efficiency (Tables 2 and 3). This study also found that maize yield and water use efficiency were not significantly different between ridges mulching with liquid and biodegradable plastic film and furrows with or

without mulching, but both were significantly higher than the without mulching treatment. The maize yield was significantly higher in the mulching black plastic film furrow with straw than in the furrow-without-straw treatment, probably due to the fact that the black mulch has a smaller light transmission rate, which effectively inhibits the evaporation of soil moisture and promotes the growth of the crop, which in turn increases the maize yield. However, based on long-term environmental impact considerations, ridges mulched with liquid plastic film may be a very promising agricultural management system, while improving maize yields and water use efficiency.

## 5. Conclusions

The maize plant height, stem thickness, leaf area and aboveground dry matter of the YJ, YL, HJ, HL, SJ and SL treatments were significantly higher than those of the conventional NJ and CK treatments, and at the same time had a better stabilizing effect on soil water content and temperature. During the complete reproductive period of maize, soil water content under different plastic film mulching treatments of dryland maize showed larger mean values at seedling and elongation stage, which were located in the range of 16–20%. The maximum values of maize yield and water use efficiency were recorded for HJ treatment with maximum values of 9613.50 kg ha$^{-1}$ and 26.33 kg ha$^{-1}$ mm$^{-1}$, respectively. The water use efficiency under different mulching methods was shown as follows: black plastic film > biodegradable plastic film > liquid plastic film > without mulching. The best mulching treatment obtained by the entropy AHP-TOPSIS evaluation methods were YL and HJ. According to the results of the study, the planting method of mulching straw in the furrow with black plastic film on the ridge has the highest water use efficiency, and in order to improve the quality of the environment, mulching liquid plastic film (YL) could also be chosen to plant dryland maize. Therefore, in order to further optimize the film mulching method of dryland agriculture and improve the yield and water use efficiency of dryland maize in the Loess Plateau, we are carrying out continuous years of positioning experiments.

## Author Contributions

**Data curation:** Hongjuan Zhang, Yunpeng Xing, Lian Xue.

**Funding acquisition:** Rui Zhang.

**Investigation:** Yunpeng Xing.

**Methodology:** Lian Xue.

**Writing – original draft:** Hongjuan Zhang.

**Writing – review & editing:** Rui Zhang.

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
