## [Decision Letter · Decision Letter 0]

8 May 2024

PONE-D-24-10502Optimizing plastic mulching to improve maize yield and water use efficiency in Loess Plateau, ChinaPLOS ONE

Dear Dr. Zhang,

Thank you for submitting your manuscript to PLOS ONE. After careful consideration, we feel that it has merit but does not fully meet PLOS ONE’s publication criteria as it currently stands. Therefore, we invite you to submit a revised version of the manuscript that addresses the points raised during the review process.

**A better title of the paper can be thought of. **

We look forward to receiving your revised manuscript.

Kind regards,

Debarati Bhaduri, Ph.D.

Academic Editor

PLOS ONE

“This research has been supported by the Fuxi youth talent training program” of Gansu Agricultural University (Gaufx-03Y10), the construction project of the "Innovative Team for Water Saving Irrigation and Water Resource Regulation in Arid Irrigation Areas" in the discipline of water conservancy engineering at Gansu Agricultural University (2023-38) and the Gansu Provincial Water Conservancy Science Experimental Research and Technology Promotion Program project (23GSLK047).”

“We thank the “Fuxi youth talent training program” of Gansu Agricultural University (Gaufx-03Y10), the construction project of the "Innovative Team for Water Saving Irrigation and Water Resource Regulation in Arid Irrigation Areas" in the discipline of water conservancy engineering at Gansu Agricultural University (2023-38) and the Gansu Provincial Water Conservancy Science Experimental Research and Technology Promotion Program project (23GSLK047) for providing financial support.”

“This research has been supported by the Fuxi youth talent training program” of Gansu Agricultural University (Gaufx-03Y10), the construction project of the "Innovative Team for Water Saving Irrigation and Water Resource Regulation in Arid Irrigation Areas" in the discipline of water conservancy engineering at Gansu Agricultural University (2023-38) and the Gansu Provincial Water Conservancy Science Experimental Research and Technology Promotion Program project (23GSLK047).”

Additional Editor Comments:

Authors are requested to follow the detailed comments provided by Reviewer 1 and Reviewer 2, while each response should be recorded in a formal author response letter.

A better title can be opted.

Reviewers' comments:

Reviewer's Responses to Questions

**Comments to the Author**

1. Is the manuscript technically sound, and do the data support the conclusions?

Reviewer #1: Yes

Reviewer #2: No

2. Has the statistical analysis been performed appropriately and rigorously? 

Reviewer #1: Yes

Reviewer #2: No

3. Have the authors made all data underlying the findings in their manuscript fully available?

Reviewer #1: Yes

Reviewer #2: Yes

4. Is the manuscript presented in an intelligible fashion and written in standard English?

Reviewer #1: Yes

Reviewer #2: No

5. Review Comments to the Author

Reviewer #1: Dear Authors

Congratulations for nice peace of research on "Optimizing plastic mulching to improve maize yield and water use efficiency in Loess Plateau, China".

The manuscript provides comparable scientific results for soil moisture, temperature, growth and yield attributes of maize crop, water use efficiency and economic returns as influenced by different methods of mulching with and without straw both in flat bed as well as ridge and furrow land configuration. All these mulching methods were also compared with ridge-furrow uncovered and traditional flat bed planting methods.

The manuscript contains all the relevant technical information and conclusions are being supported with sufficient data with statistical analysis which were presented intelligently in whole manuscript.

However author needs to be verified some of the technical writings which needs significant improvement in the manuscript viz., soil evaporation (it may soil moisture evaporation), maize water (It may be the water in maize crop), rained (it may be rain fed) etc.

Following sentence lines may be verified in the edited version of manuscript:

Line 31: Soil evaporation

Line 33: The soil temperature and increase in yield of maize.

Line 34: Add suitable reference in support of the statement made.

Line 40: study of efficient water resource utilization in maize crop.

Line 41: maize yield

Line 47: methods were investigated

Line 57: Soil properties were pH.....

Line 80: Soil moisture (suitable Ref.)

Line 89: What is SWS abbreviate the same

Line 91: Soil temperature (Ref. if any)

Line 97: plant fixation observation.. clarify the fixation word here

Line 125: rained (it may be raifed)

Line 299: Connect and continue the sentence

The author may kindly correct the whole document in the light or errors enumerated above. Further the data presented in the manuscript is for only one season which has to be reproduced in the similar set of experiment for one more season for consistency of the information's generated. Therefore manuscript may be supported with data of two cropping seasons.

Reviewer #2: The research entitled “Optimizing plastic mulching to improve maize yield and water use efficiency in Loess Plateau, China” highlighting the significance of in-situ water and its impact on soil moisture and temperature regime as well as crop response. The study is well thought and designed. Overall manuscript is good, however here are some of critical miss in the present version of manuscript. Most importantly, message from author is tough to understand which need to revisit.

Abstract:

• Different mulching practices “of maize growth”, soil should be “On maize growth” [line7]

• Rewrite the sentence “The results showed that compared with CK and NJ, the soil moisture and soil moisture………… [line 11-12]

• Specify the Black mulch, is it black plastic? Mulch straw of which crop/biomass?

• What was the liquid used as “Liquid mulch”? Specify

• Rewrite the sentence “Application to the 2019 maize of 830 kg ha-1 (compound fertilizer, N-P2O5- K2O5=18-12-10) and 7,500 kg ha-1 (organic fertilizer) before sowing of maize in 2019, and without irrigation throughout the reproductive period” [line – 73-75].

• What author want to highlight not clear in the paragraph from line 72 – 78, rewrite.

Introduction: okay

Materials and Methods: Appropriate

• Given the fact that no external water was supplied to the crop, soil water holding capacity and moisture content at FC and PWP should be given for particular soil.

• Detailed methodological description is required for APH-TOPSIS models for better understanding.

Result and Discussion:

• Include the nutrient profile of Milk Vetch which was incorporated in the pot experiments.

• The title of figure-5 is not appropriate, as it is representing the parodic soil moisture pattern instead of daily moisture content. May be revised

• There is no preceding crop mention in the manuscript but moisture storage at pre-planting differs significantly while it was non-significant at post-planting despite the higher standard deviation (figure-6). Given the fact that, crop moisture drawing is supposed to differ significantly due to contrasting difference in the mulching type and source. Proper justification may be incorporated.

• LSD value or alphabetic depicting missing in figure -9.

• Result interpretation for APH-TOPSIS is improper. Need to rephased.

• Moreover, Discussion part if very constrained which may be even more elaborative correlating the findings of present experiment.

Conclusion: A future implication statement is missing.

6. PLOS authors have the option to publish the peer review history of their article (what does this mean?). If published, this will include your full peer review and any attached files.

Reviewer #1: No

Reviewer #2: **Yes: **Shiv Vendra Singh

---

## [Author Response · Author response to Decision Letter 0]

24 Jun 2024

Dear Editor and Reviewers, Our response comments are uploaded in the attached document.

---

## [Decision Letter · Decision Letter 1]

16 Jul 2024

PONE-D-24-10502R1Optimizing plastic film mulch to improve the yield and water use efficiency of dryland maize in the Loess Plateau, ChinaPLOS ONE

Dear Dr. Zhang,

Thank you for submitting your manuscript to PLOS ONE. After careful consideration, we feel that it has merit but does not fully meet PLOS ONE’s publication criteria as it currently stands. Therefore, we invite you to submit a revised version of the manuscript that addresses the points raised during the review process.

We look forward to receiving your revised manuscript.

Kind regards,

Debarati Bhaduri, Ph.D.

Academic Editor

PLOS ONE

Journal Requirements:

Additional Editor Comments:

I must advise to check these as one of the reviewers pointed out-

• For economic assessment, cost of cultivation and per unit price/cost should be given.

• Is there any significance of including alphabets in rainfall data as it is not a variable (table 3).

• Dryland harvesting treatment pattern in line 453-455 does not make sense, please specify the meaning!

Point-to-point response to comments by Reviewer 1 should be provided.

Reviewers' comments:

Reviewer's Responses to Questions

**Comments to the Author**

1. If the authors have adequately addressed your comments raised in a previous round of review and you feel that this manuscript is now acceptable for publication, you may indicate that here to bypass the “Comments to the Author” section, enter your conflict of interest statement in the “Confidential to Editor” section, and submit your "Accept" recommendation.

Reviewer #1: (No Response)

Reviewer #2: All comments have been addressed

2. Is the manuscript technically sound, and do the data support the conclusions?

Reviewer #1: Partly

Reviewer #2: Yes

3. Has the statistical analysis been performed appropriately and rigorously? 

Reviewer #1: Yes

Reviewer #2: Yes

4. Have the authors made all data underlying the findings in their manuscript fully available?

Reviewer #1: (No Response)

Reviewer #2: Yes

5. Is the manuscript presented in an intelligible fashion and written in standard English?

Reviewer #1: Yes

Reviewer #2: Yes

6. Review Comments to the Author

Reviewer #1: The manuscript contains all the relevant technical information produced scientifically but the conclusions made needs to be verified for the reproducibility of the information which was also requested in the first reviewer report. For this authors may include the experimental results for multiple seasons/years of experimentation and resubmit the manuscript for consideration.

Reviewer #2: Authors have addressed the queries raised in the submitted manuscript. Revised manuscript is well structured compared to previous version. Manuscript can be accepted after minor inclusions. I shall suggest to critically check the language.

• For economic assessment, cost of cultivation and per unit price/cost should be given.

• Is there any significance of including alphabets in rainfall data as it is not a variable (table 3).

• Dryland harvesting treatment pattern in line 453-455 does not make sense, please specify the meaning!

7. PLOS authors have the option to publish the peer review history of their article (what does this mean?). If published, this will include your full peer review and any attached files.

Reviewer #1: No

Reviewer #2: **Yes: **Shiv Vendra Singh

---

## [Author Response · Author response to Decision Letter 1]

23 Jul 2024

We have uploaded the response to the reviewers' comments as an attachment.

---

## [Decision Letter · Decision Letter 2]

30 Jul 2024

Optimizing plastic film mulch to improve the yield and water use efficiency of dryland maize in the Loess Plateau, China

PONE-D-24-10502R2

Dear Dr. Zhang,

We’re pleased to inform you that your manuscript has been judged scientifically suitable for publication and will be formally accepted for publication once it meets all outstanding technical requirements.

Kind regards,

Debarati Bhaduri, Ph.D.

Academic Editor

PLOS ONE

Additional Editor Comments (optional):

Reviewers' comments:

Reviewer's Responses to Questions

**Comments to the Author**

1. If the authors have adequately addressed your comments raised in a previous round of review and you feel that this manuscript is now acceptable for publication, you may indicate that here to bypass the “Comments to the Author” section, enter your conflict of interest statement in the “Confidential to Editor” section, and submit your "Accept" recommendation.

Reviewer #2: All comments have been addressed

2. Is the manuscript technically sound, and do the data support the conclusions?

Reviewer #2: Yes

3. Has the statistical analysis been performed appropriately and rigorously? 

Reviewer #2: Yes

4. Have the authors made all data underlying the findings in their manuscript fully available?

Reviewer #2: (No Response)

5. Is the manuscript presented in an intelligible fashion and written in standard English?

Reviewer #2: (No Response)

6. Review Comments to the Author

Reviewer #2: The authors have addressed all the comments, and the revised manuscript is scientifically sound to be considered.

7. PLOS authors have the option to publish the peer review history of their article (what does this mean?). If published, this will include your full peer review and any attached files.

Reviewer #2: **Yes: **Shiv Vendra Singh

---

## [Editor Report · Acceptance letter]

10 Aug 2024

PONE-D-24-10502R2 

PLOS ONE

Dear Dr. Zhang, 

I'm pleased to inform you that your manuscript has been deemed suitable for publication in PLOS ONE. Congratulations! Your manuscript is now being handed over to our production team.

Kind regards, 

on behalf of

Dr. Debarati Bhaduri 

Academic Editor

PLOS ONE